# Clinical characteristics and outcomes of critically ill patients with COVID-19 admitted to an intensive care unit in London: A prospective observational cohort study

Ross J. Thomson[ORCID][1,2]*, Jennifer Hunter[3], Jonathan Dutton[3], James Schneider[3], Maryam Khosravi[1,4], Alisha Casement[1], Kulwant Dhadwal[1,3], Daniel Martin[1,5]

1 Department of Intensive Care Medicine, Royal Free Hospital, Royal Free London NHS Foundation Trust, London, United Kingdom, 2 William Harvey Research Institute, Queen Mary, University of London, London, United Kingdom, 3 Department of Anaesthesia, Royal Free Hospital, Royal Free London NHS Foundation Trust, London, United Kingdom, 4 Department of Nephrology, Royal Free Hospital, Royal Free London NHS Foundation Trust, London, United Kingdom, 5 Peninsula Medical School, University of Plymouth, Plymouth, United Kingdom

* ross.thomson@nhs.net

**Data Availability Statement:** The data that support the findings of this study are not publicly available out of consideration for patient confidentiality. On

## Abstract

### Background

Cohorts of severely ill patients with COVID-19 have been described in several countries around the globe, but to date there have been few published reports from the United Kingdom (UK). Understanding the characteristics of the affected population admitted to intensive care units (ICUs) in the UK is crucial to inform clinical decision making, research and planning for future waves of infection.

### Methods

We conducted a prospective observational cohort study of all patients with COVID-19 admitted to a large UK ICU from March to May 2020 with follow-up to June 2020. Data were collected from health records using a standardised template. We used multivariable logistic regression to analyse the factors associated with ICU survival.

### Results

Of the 156 patients included, 112 (72%) were male, 89 (57%) were overweight or obese, 68 (44%) were from ethnic minorities, and 89 (57%) were aged over 60 years of age. 136 (87%) received mechanical ventilation, 77 (57% of those intubated) were placed in the prone position and 95 (70% of those intubated) received neuromuscular blockade. 154 (99%) patients required cardiovascular support and 44 (28%) required renal replacement therapy. Of the 130 patients with completed ICU episodes, 38 (29%) died and 92 (71%) were discharged alive from ICU. In multivariable models, age (OR 1.13 [95% CI 1.07–1.21]), obesity (OR 3.06 [95% CI 1.16–8.74]), lowest P/F ratio on the first day of admission (OR 0.82 [95% CI 0.67–0.98]) and $PaCO_2$ (OR 1.52 [95% CI 1.01–2.39]) were independently associated with ICU death.

account of the highly granular nature of the clinical dataset it is not possible to anonymise it sufficiently to prevent reidentification of participants without jeopardising its usefulness for research. The Institutional Research Office (rf.randd@nhs.net) can facilitate access to the data for appropriately qualified individuals provided that the correct ethical permissions and data sharing agreements are in place.

**Funding:** The authors received no specific funding for this work.

**Competing interests:** I have read the journal's policy and the authors of this manuscript have the following competing interests: DM has received consultancy and lecture fees from Edwards Lifesciences and Siemens Healthineers; none of the other authors have any conflicts of interest to report. This does not alter our adherence to PLOS ONE policies on sharing data and materials.

## Conclusions

Age, obesity and severity of respiratory failure were key determinants of survival in this cohort. Multiorgan failure was prevalent. These findings are important for guiding future research and should be taken into consideration during future healthcare planning in the UK.

## Introduction

The global pandemic of coronavirus disease 2019 (COVID-19), the illness caused by infection with severe acute respiratory syndrome coronavirus 2 (SARS-CoV-2), has affected tens of millions of people and led to over one million deaths [1]. The proportion of patients with severe illness requiring admission to an intensive care unit (ICU) has been reported at between 4% and 32% [2], and concerns that ICU capacity may be overwhelmed have weighed heavily in policy considerations such as the implementation of lockdowns and social distancing [3].

Cohorts of patients critically ill with COVID-19 have been described by authors from several countries, including China [4, 5], Italy [6], Sweden [7] and the United States [8–10]. From these studies we have learnt important lessons including the preponderance of males being affected, the association of increasing age with mortality, and the high prevalence of co-morbidities such as hypertension, diabetes and obesity. Patients most severely affected by COVID-19 are likely to be admitted to an ICU; understanding the demographic pattern of these patients and factors related to important clinical outcomes is essential. To date, peer-reviewed analysis of such patients in the United Kingdom (UK) has been limited to large scale epidemiological studies or focussed studies in small samples. We therefore conducted a prospective observational cohort study to better understand the clinical characteristics and outcomes of patients admitted to an ICU in the UK with severe COVID-19. Detailed analysis of this cohort is vital to gain insight into the factors associated with outcomes, guide planning for future waves of infection, and to inform clinical decision making and research.

## Methods

### Study design and participants

We performed a prospective observational cohort study at the Royal Free Hospital [11], a 520 bed teaching hospital in London, UK. The Royal Free Hospital is one of four designated centres for managing patients with airborne high consequence infectious diseases in the UK [12] and was the second hospital in the country to admit a patient with confirmed COVID-19. We enrolled all patients with laboratory confirmed SARS-CoV-2 infection admitted to the ICU from the first case until the cut-off date for this study, 6 May 2020. This date was chosen because there were no further ICU admissions in the subsequent two weeks. Patients were identified by daily review of the ICU admission database. Follow-up was right-censored on 10 June 2020, giving at least 28 days' follow-up in every patient. The initial capacity of the ICU was 34 patients; this was scaled up to 70 patients at the height of the pandemic.

A standard operating procedure for identification of patients requiring admission to the ICU was devised in line with the WHO guidance on the management of patients with COVID-19 [13]. Patients with critical COVID-19 infection, defined as the presence of ARDS, sepsis or septic shock, were admitted to the ICU unless this was contraindicated. Patients with severe COVID-19 infection, defined as respiratory rate > 30 breaths/min; severe respiratory distress; or $SpO_2 < 90\%$ on room air, were kept under close observation. In line with guidance

issued by the UK National Institute for Health and Care Excellence [14], the Clinical Frailty Score was calculated for every patient admitted to hospital. This, together with a holistic assessment of each patient's condition, including their comorbidities, physiological reserve and their wishes and those of their families, were used to determine when admission to the ICU was likely to be futile. There were no exclusion criteria for the study and there was no sample size calculation; the size of the cohort was determined by the number of patients admitted during the study period.

Diagnosis of SARS-CoV-2 infection was made using RT-PCR of nasopharyngeal secretions, sputum or endotracheal aspirate. At the beginning of the pandemic all samples were sent to a regional reference laboratory operated by Public Health England; subsequently an in-house assay was developed and this was later supplemented by commercial assays.

The study was classified as a non-interventional service evaluation using routinely collected patient data and was registered with the institutional audit department. The UK Policy Framework for Health and Social Care does not require ethical approval or explicit patient consent for such studies.

## Procedures

We captured routinely collected patient data from paper-based and electronic health records using a standardised template derived from the International Severe Acute Respiratory and emerging Infection Consortium (ISARIC) case report form [15] together with additional variables hypothesised to be relevant, based on the published literature at the start of the study period. The dataset consisted of demographic characteristics (age, sex, self-reported ethnicity and body mass index [BMI]), comorbidities (hypertension, hyperlipidaemia, diabetes, ischaemic heart disease, chronic respiratory disease, smoking status, chronic kidney disease, end-stage renal failure [ESRF] requiring renal replacement therapy), details of the presenting illness including the nature of symptoms and their duration, the initial hospital course prior to ICU admission, physiological variables on hospital and ICU admission and on days 1, 3 and 7 of the ICU admission, details of treatments received on ICU and pathology and radiology reports. We classified cardiovascular and respiratory support according to the definitions used by the UK Intensive Care National Audit and Research Centre [16].

## Statistical analysis

We analysed the data using R version 4.0.0 with RStudio version 1.3.959. All of the authors had unrestricted access to the raw data. Missing data were not imputed. Continuous variables were summarised using medians and interquartile ranges with comparisons between groups using the Wilcoxon rank-sum test. Categorical variables were presented as numbers and percentages with comparisons between groups using the chi-square or Fisher exact tests. p-values have not been adjusted to take account of multiple comparisons.

We used logistic regression to assess the factors associated with ICU survival. Only patients with completed ICU episodes (i.e. those who died on or were discharged alive from ICU, excluding those who were transferred out to other hospitals) were included in these analyses. We created two sets of models, the first employing patient characteristics and physiology on admission to ICU, and the second using physiology, treatments and complications during the ICU admission. We captured each patient's most extreme physiological variables on days 1, 3 and 7 of the ICU admission. For each model set we performed univariable regressions using variables thought to be associated with survival based on the published literature and clinical experience. From these univariable models we selected those variables found to have a statistically significant association with outcome at the $p < 0.1$ level and included them in a

multivariable model. For each variable we presented the (adjusted) odds ratio for death together with the associated 95% confidence interval and p value.

## Results

### Baseline characteristics

Between 2 March and 6 May 2020, 156 patients were admitted to our ICU with COVID-19. 112 (72%) were male, the median (IQR) age was 62 (54 to 70) years and 89 (57%) patients were aged over 60 years. The majority of the patients (89 [57%]) were overweight or obese (BMI $\geq$ 25 kg/m$^2$). With regards to ethnicity, 36 (23%) were Asian and 32 (21%) were Black. 26 patients (17%) had no reported past medical history. The most common comorbidities were hypertension (81 [52%]), dyslipidaemia (56 [36%]) and diabetes mellitus (52 [33%]). Baseline demographic characteristics of the cohort are shown in Table 1 and comorbidities are shown in Fig 1. The number of admissions, discharges, transfers and deaths over time are show in Fig 2. By way of context, 738 patients with a laboratory-confirmed diagnosis of COVID-19 were admitted to the Royal Free Hospital over the same time period. Of the 582 who were not admitted to ICU, the median (IQR) age was 74 (59 to 85) years and 421 (72%) were aged over 60 years. The data for the hospital were derived from an administrative database that did not record clinical characteristics.

Patients reported, on average, a one week history of symptoms at the time of hospital admission (median 7 days, IQR 5 to 10 days). The most common symptoms at the time of admission were breathlessness (127 [81%]), cough (125 [81%]) and fever (122 [78%]). The range of symptoms on admission is presented in Fig 3. 109 (70%) patients were initially

**Table 1. Baseline characteristics of the population.**

| Characteristic | N = 156[1] |
|---|---|
| **Gender** | |
| Female | 44 (28%) |
| Male | 112 (72%) |
| **Age** | 62 (54, 70) |
| **Age Group** | |
| Under 20 | 0 (0%) |
| 20 to 40 | 8 (5.1%) |
| 40 to 60 | 59 (38%) |
| 60 to 80 | 86 (55%) |
| 80+ | 3 (1.9%) |
| **Ethnicity** | |
| White | 73 (47%) |
| Black | 32 (21%) |
| Asian | 36 (23%) |
| Other | 15 (9.6%) |
| **BMI Group** | |
| Under 18.5 | 0 (0%) |
| 18.5 to 25 | 67 (43%) |
| 25 to 30 | 58 (37%) |
| 30 to 40 | 20 (13%) |
| 40+ | 11 (7.1%) |

[1]Statistics presented: n (%); median (IQR)

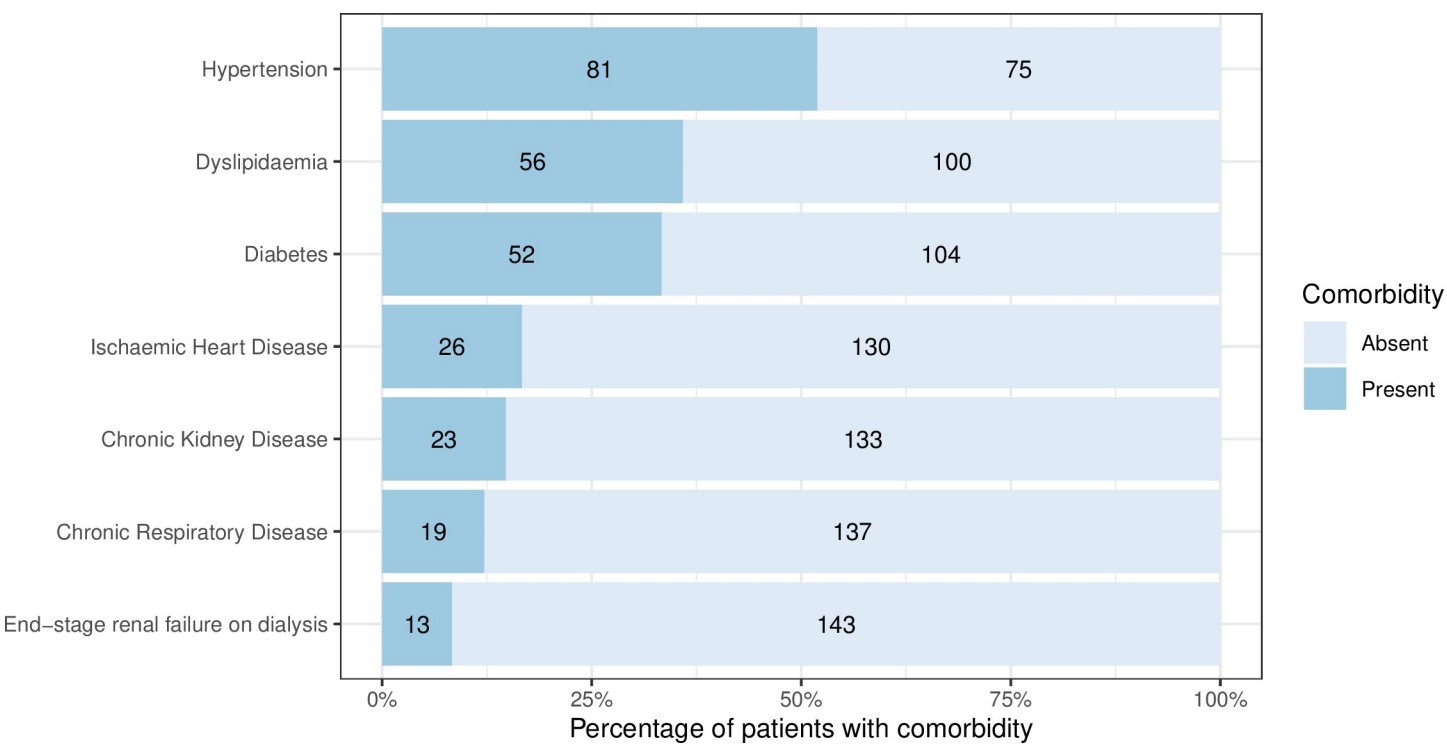

**Fig 1. Comorbidities at hospital admission.**

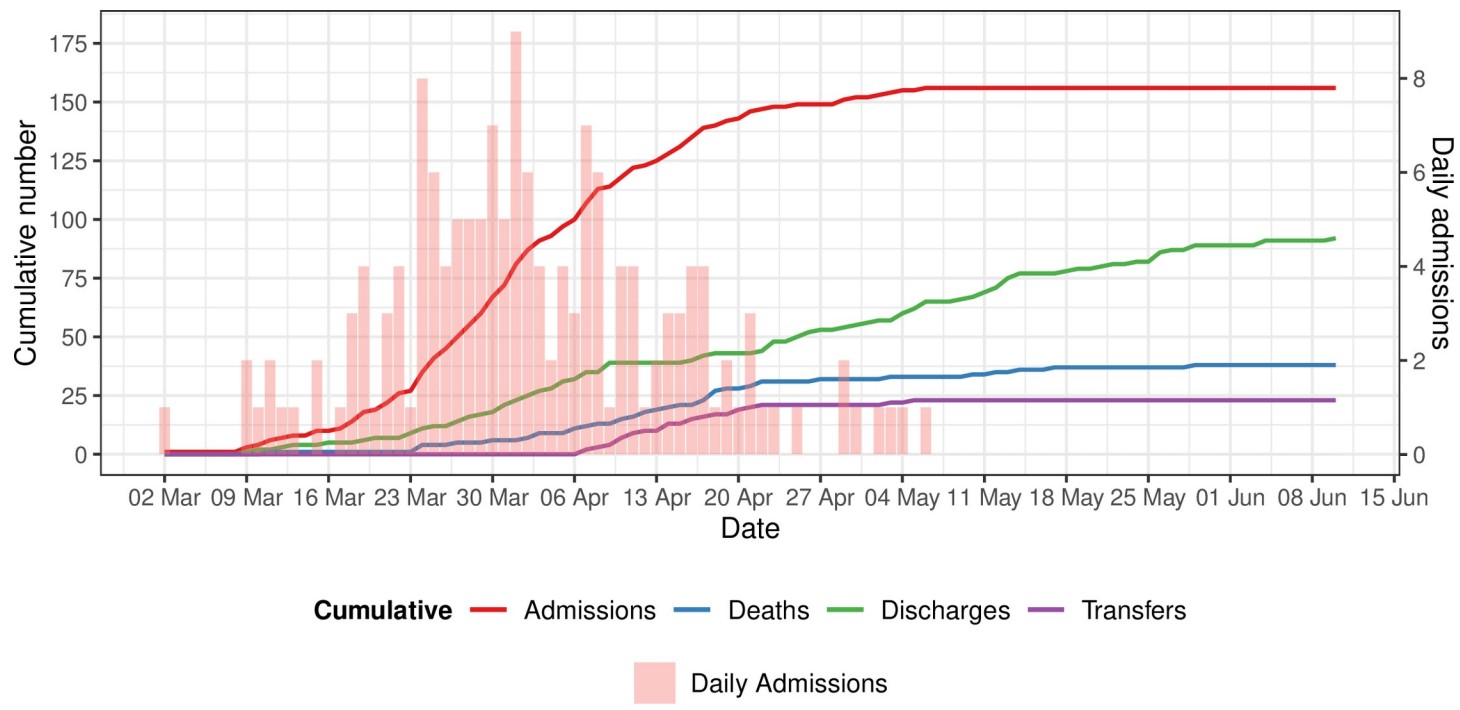

**Fig 2. Admissions, discharges, transfers and death over time.**

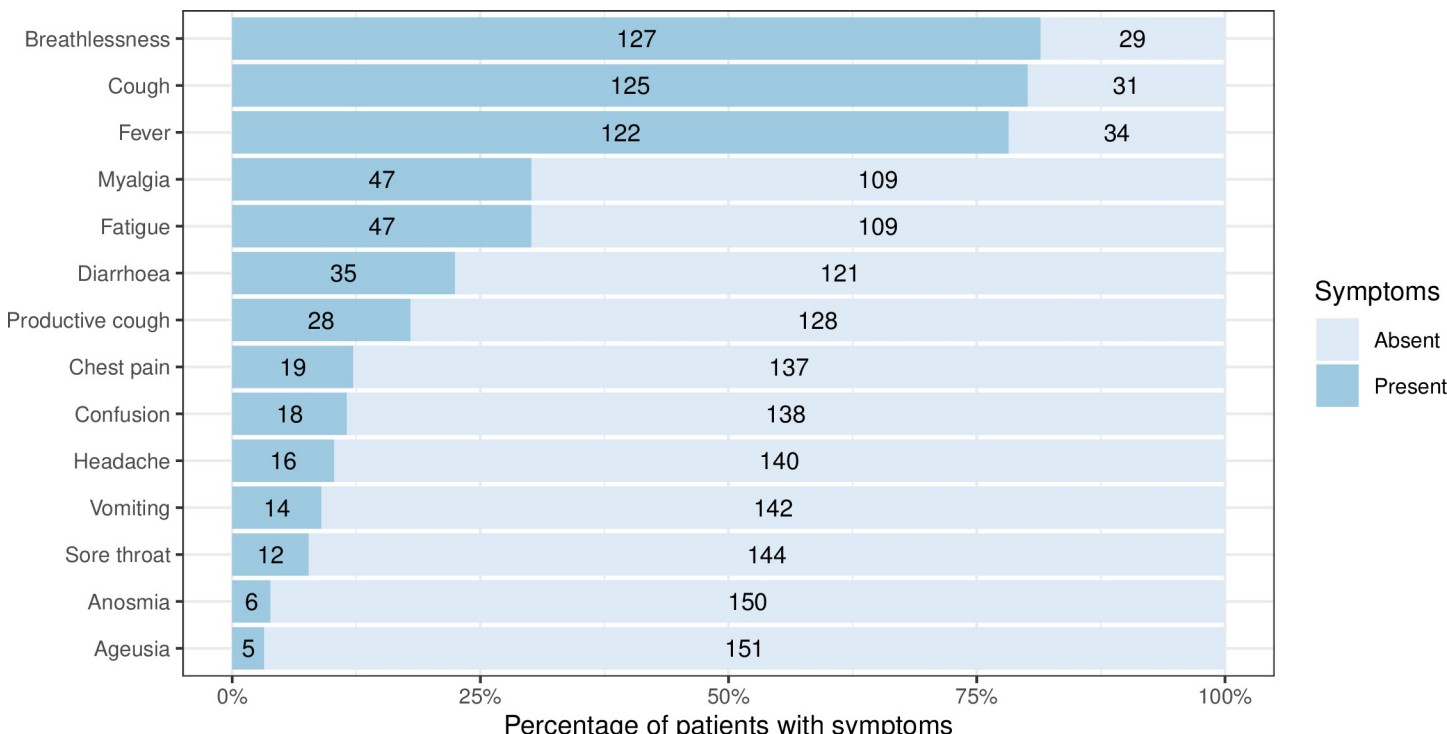

**Fig 3. Symptoms at hospital admission.**

admitted to a ward. For these patients, the median (IQR) length of stay prior to ICU admission was 55 (29 to 87) hours.

## Physiology

Patients were profoundly hypoxaemic on admission to ICU, with a median ratio of arterial partial pressure of oxygen ($PaO_2$) to inspired fraction of oxygen ($FIO2$) (P/F ratio) of 17.1 (IQR 13.2 to 21.3) kPa (approximately 125 mmHg). Compared to those patients who survived to ICU discharge, those who died had persistently lower P/F ratios (15.8 versus 17.9, p = 0.017 on day 1), lower arterial pH (7.3 versus 7.4, p = 0.031 on day 1) and higher arterial partial pressure of carbon dioxide ($PaCO_2$) (6.0 versus 5.5 kPa, p = 0.040 on day 1) on days 1, 3 and 7 of admission. Furthermore, those patients who died had higher peak inspiratory pressure (PIP) on days 3 and 7; this was predominantly driven by a reduction of PIP in the group who survived and a rise of PIP in the group who died, reflecting changes in lung compliance over time. Patients who survived had lower peak noradrenaline doses on day 3 (0.10 versus 0.15 mcg/kg/min, p = 0.030) and day 7 (0.07 versus 0.15 mcg/kg/min, p = 0.003). Patients who died had higher positive cumulative fluid balance on the third (1,962 versus 1,350 ml, p = 0.045) and seventh (4,645 versus 1,332 ml, p<0.001) days of admission compared to those who survived. There were no differences between those who died and those who survived in the lowest recorded mean arterial blood pressure or highest temperature. Physiological measures for patients with completed ICU episodes are presented in Table 2.

## Treatments received on ICU

136 (87%) patients were intubated for mechanical ventilation during their ICU admission, with this occurring less than one hour after ICU admission in 104 (67%) patients. 77 (57% of

**Table 2. Physiological measurements over time, stratified by ICU survival.**

| Characteristic | Overall, N = 156 | Died, N = 38[1] | Surviving, N = 118[1] | p-value[2] |
|---|---|---|---|---|
| **Lowest P/F Ratio** | | | | |
| Day 1 | 17.1 (13.2 to 21.3) | 15.8 (12.1 to 18.3) | 17.9 (13.6 to 22.3) | 0.017 |
| Day 3 | 17.7 (13.9 to 23.6) | 16.1 (12.7 to 18.7) | 18.2 (14.4 to 24.6) | 0.006 |
| Day 7 | 17.6 (13.6 to 23.2) | 12.9 (10.1 to 16.3) | 19.2 (15.8 to 24.2) | <0.001 |
| **pH at the time of lowest P/F Ratio** | | | | |
| Day 1 | 7.4 (7.3 to 7.4) | 7.3 (7.3 to 7.4) | 7.4 (7.3 to 7.4) | 0.031 |
| Day 3 | 7.4 (7.3 to 7.4) | 7.3 (7.3 to 7.4) | 7.4 (7.4 to 7.4) | <0.001 |
| Day 7 | 7.4 (7.3 to 7.4) | 7.4 (7.3 to 7.4) | 7.4 (7.4 to 7.5) | <0.001 |
| **PaCO$_2$ at the time of the lowest P/F Ratio (kPa)** | | | | |
| Day 1 | 5.7 (5.1 to 6.5) | 6.0 (5.3 to 6.5) | 5.5 (5.0 to 6.4) | 0.040 |
| Day 3 | 6.1 (5.4 to 6.9) | 6.8 (6.1 to 7.8) | 5.8 (5.2 to 6.6) | <0.001 |
| Day 7 | 5.9 (5.2 to 6.8) | 6.3 (5.4 to 7.2) | 5.8 (5.0 to 6.6) | 0.029 |
| **PEEP at the time of the lowest P/F ratio (cmH$_2$O)** | | | | |
| Day 1 | 10.0 (10.0 to 12.0) | 10.0 (10.0 to 12.5) | 10.0 (10.0 to 12.0) | 0.5 |
| Day 3 | 10.0 (9.0 to 12.0) | 12.0 (10.0 to 12.5) | 10.0 (8.0 to 12.0) | 0.056 |
| Day 7 | 10.0 (8.0 to 12.0) | 10.0 (8.2 to 12.4) | 10.0 (8.0 to 12.0) | 0.055 |
| **PIP at the time of the lowest P/F Ratio (cmH$_2$O)** | | | | |
| Day 1 | 27.0 (23.0 to 29.0) | 27.0 (24.0 to 29.0) | 26.0 (23.0 to 29.0) | 0.4 |
| Day 3 | 26.0 (21.8 to 29.0) | 27.0 (24.2 to 30.0) | 26.0 (21.0 to 28.0) | 0.034 |
| Day 7 | 25.0 (20.0 to 30.0) | 30.0 (25.0 to 33.0) | 24.0 (18.8 to 28.0) | <0.001 |
| **Cumulative fluid balance in 24 hours (ml)** | | | | |
| Day 1 | 648 (99 to 1334) | 850 (450 to 1386) | 600 (4 to 1220) | 0.062 |
| Day 3 | 1700 (318 to 3022) | 1962 (1250 to 3620) | 1350 (-22 to 2590) | 0.045 |
| Day 7 | 1888 (56 to 4726) | 4645 (2963 to 6485) | 1332 (-309 to 3493) | <0.001 |
| **Mean arterial blood pressure (mmHg)** | | | | |
| Day 1 | 68 (63 to 75) | 68 (63 to 75) | 68 (63 to 75) | 0.9 |
| Day 3 | 68 (65 to 75) | 65 (65 to 75) | 68 (65 to 75) | 0.7 |
| Day 7 | 71 (65 to 80) | 70 (60 to 75) | 74 (65 to 80) | 0.2 |
| **Maximum noradrenaline dose in 24 hours (mcg/kg/min)** | | | | |
| Day 1 | 0.11 (0.07 to 0.17) | 0.13 (0.08 to 0.24) | 0.11 (0.07 to 0.16) | 0.2 |
| Day 3 | 0.11 (0.07 to 0.18) | 0.15 (0.09 to 0.26) | 0.10 (0.06 to 0.15) | 0.030 |
| Day 7 | 0.10 (0.05 to 0.16) | 0.15 (0.10 to 0.27) | 0.07 (0.04 to 0.13) | 0.003 |
| **Maximum temperature in 24 hours (°C)** | | | | |
| Day 1 | 38.0 (37.2 to 38.8) | 38.2 (37.4 to 38.8) | 37.9 (37.2 to 38.8) | 0.4 |
| Day 3 | 37.7 (37.1 to 38.5) | 37.8 (37.1 to 38.2) | 37.7 (37.2 to 38.5) | 0.3 |
| Day 7 | 37.5 (37.2 to 37.9) | 37.4 (37.1 to 37.9) | 37.5 (37.2 to 37.9) | 0.6 |

[1]Statistics presented: median (IQR)

[2]Statistical tests performed: Wilcoxon rank-sum test Patients who were transferred out or who were still on ICU at the time of analysis were classed as Surviving

those intubated) patients were placed in the prone position for mechanical ventilation at some point during their ICU stay, while 95 (70% of those intubated) received neuromuscular blockade (over and above that given at the time of intubation). The median (IQR) time to administration of neuromuscular blockade was 24 hours (0 to 48) and the median (IQR) time to prone positioning was 48 hours (0 to 96). 52 (38% of those intubated) patients ultimately underwent tracheostomy insertion to facilitate weaning from the ventilator; this occurred a median (IQR) of 15.8 days (12.6 to 21) after ICU admission.

**Table 3. Organ support received on ICU.**

| Characteristic | N = 156[1] |
|---|---|
| **Cardiovascular support (ICNARC definition)** | |
| Advanced | 35 (22%) |
| Basic | 119 (76%) |
| None | 2 (1.3%) |
| Missing | 0 (0%) |
| **Respiratory support (ICNARC definition)** | |
| Advanced | 141 (90%) |
| Basic | 15 (9.6%) |
| None | 0 (0%) |
| Missing | 0 (0%) |
| **Renal replacement therapy** | 44 (28%) |
| **Number of days of renal replacement therapy** | 8 (4 to 22) |

[1]Statistics presented: n (%); median (25% to 75%)

The majority of patients admitted to ICU required organ support in addition to mechanical ventilation. 119 (76%) patients required a single vasopressor drug while 35 (23%) patients required multiple vasopressor or inotropic medications. 44 (28%) patients required renal replacement therapy (continuous venovenous haemofiltration or haemodialysis), for a median (IQR) duration of 8 (4 to 22) days. All patients received broad-spectrum antibiotics for the empirical treatment of super-added bacterial pneumonia. Details of organ support are presented in Table 3.

8 (5.1%) patients were enrolled in a randomised control trial of remdesivir versus placebo (clinicaltrials.gov registration number NCT04292899) and 15 (9.6%) patients were enrolled in the COVACTA trial of tocilizumab versus placebo (clinicaltrials.gov registration number NCT04320615).

## Thromboembolic complications

82 (53%) patients underwent clinically indicated computed tomography pulmonary angiography (CTPA) to diagnose or exclude pulmonary thromboembolism (PE). Criteria for CTPA included hypoxaemia out of keeping with the appearance of the lung fields on chest radiography, extremely high D-dimer or a D-dimer that rose or remained static despite improvement of other inflammatory markers, failure to improve despite 48 hours' prone position ventilation, new onset dysrhythmia, or evidence of right heart strain on ECG or echocardiography. 44 patients (54% of those who underwent CTPA) were diagnosed with PE; the majority of these were lobar or segmental. Right heart strain was present in 15 patients (33% of those who underwent CTPA). Thromboembolic complications are presented in Table 4.

## Outcomes

Of the 156 total admissions to ICU with COVID-19, 38 (24%) patients died on ICU, 23 (15%) patients were transferred out to other hospitals, 92 (59%) patients were discharged alive from ICU and the remaining 3 (2%) patients were still on ICU at the time of follow-up. Of the 23 patients transferred out, one patient was transferred to the regional referral centre for extracorporeal membrane oxygenation and 22 patients were sent to other hospitals to balance patient capacity in London. Of the 92 patients discharged from ICU 82 (89%) were subsequently

**Table 4. Thromboembolic complications.**

| Characteristic | N = 156[1] |
|---|---|
| **CTPA performed** | 82 (53%) |
| **PE diagnosed on CTPA** | 44 (54%) |
| **Level of PE** | |
| Pulmonary trunk | 6 (14%) |
| Lobar | 10 (23%) |
| Segmental | 22 (50%) |
| Subsegmental | 6 (14%) |
| **RV strain on CTPA** | 15 (33%) |

RV = Right ventricular

Percentages are of the parent group

[1]Statistics presented: n (%)

discharged from hospital and one died on the ward. Considering all patients, including those transferred out, 116 (74%) patients survived to 30 days following ICU admission. Survival, stratified by age group and sex, is shown in Fig 4. The 23 patients transferred out to other hospitals and 3 patients still on ICU have been excluded from the analysis of outcomes. Of the 130 patients with completed ICU episodes, who were included in the logistic regression models, 92 (71%) patients survived and the median length of stay was 11.8 (6.6 to 28.7) days.

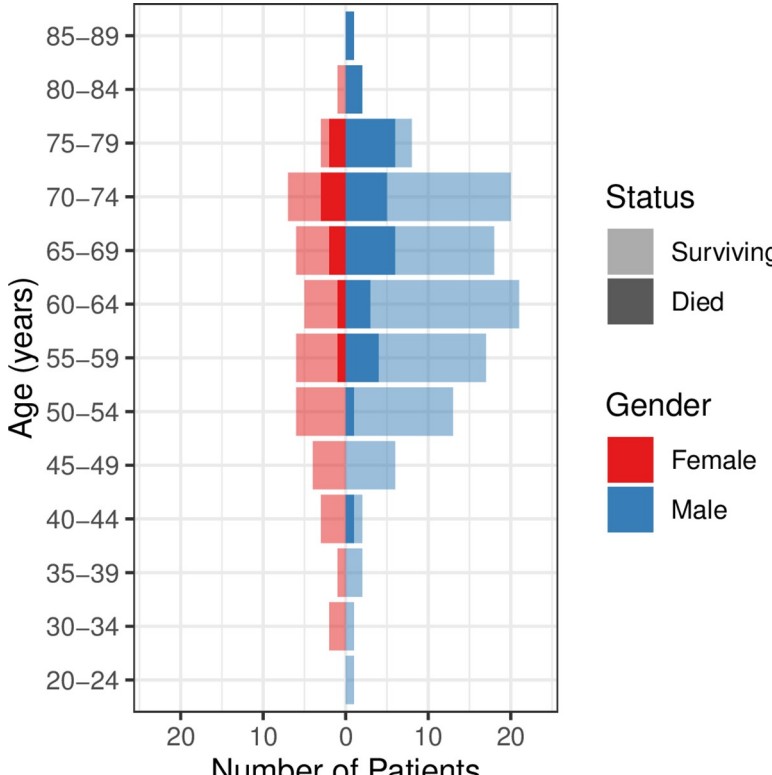

**Fig 4. Survival stratified by age and sex.**

In the first set of logistic regression models (Table 5), that employed patient characteristics on admission to ICU, age (OR 1.12 [95% CI 1.07–1.18]), Asian ethnicity (OR 2.57 [95% CI 1.02–6.57]), overweight or obese BMI (OR 1.90 [95% CI 0.87–4.33]), lowest P/F ratio on the first day of admission (OR 0.91 [95% CI 0.84–0.97]) and $PaCO_2$ at the time of the lowest P/F ratio on the first day of admission (OR 1.40 [95% CI 1.02–1.95]) were associated with increased odds of death in univariable regression models, at a significance level of p < 0.1. Arterial pH at the time of the lowest P/F ratio on the first day of admission was significantly associated with death in the statistical sense although the effect size was negligible. In a multivariable model age (OR 1.13 [95% CI 1.07–1.21]), obesity (OR 3.06 [95% CI 1.16–8.74]), lowest P/F ratio on the first day of admission (OR 0.90 [95% CI 0.81–0.98]) and $PaCO_2$ (OR 1.52 [95% CI 1.01–2.39]) remained significant at the p < 0.05 level.

In the second set of logistic regression models (Table 6), that evaluated events during ICU admission, age (OR 1.12 [95% CI 1.07–1.18]), lowest P/F ratio across days 1, 3 and of ICU admission (OR 0.80 [95% CI 0.71–0.88]), highest $PaCO_2$ across days 1, 3 and of ICU admission (OR 2.00 [95% CI 1.43–2.89]), highest positive end-expiratory pressure (PEEP) across days 1, 3 and of ICU admission (OR 1.15 [95% CI 0.99–1.35]), highest peak inspiratory pressure (PIP) across days 1, 3 and of ICU admission (OR 1.15 [95% CI 1.04–1.29]), peak noradrenaline dose across days 1, 3 and of ICU admission (OR 32.2 [95% CI 3.97–341]) and receiving neuromuscular blockade (OR 5.82 [95% CI 2.36–16.6]) or receiving prone position ventilation (OR 3.37 [95% CI 1.54–7.73]) were associated with increased odds of death in univariable models, at a significance level of p < 0.1. In a multivariable model age (OR 1.17 [95% CI 1.09–1.27]), lowest P/F ratio (OR 0.82 [95% CI 0.67–0.98]) and peak noradrenaline dose (OR 33.0 [95% CI 1.61–860]) remained significantly associated with death at the p < 0.05 level.

**Table 5. Relationships between factors on admission to ICU and outcome.**

| Characteristic | N | Univariable | | | Multivariable | | |
|---|---|---|---|---|---|---|---|
| | | OR[1] | 95% CI[1] | p-value | OR[1] | 95% CI[1] | p-value |
| Age on admission | 130 | 1.12 | 1.07, 1.18 | <0.001 | 1.13 | 1.07, 1.21 | <0.001 |
| Gender | 130 | | | | | | |
| Female | | | | | | | |
| Male | | 1.27 | 0.54, 3.17 | 0.6 | | | |
| Ethnicity | 130 | | | | | | |
| White | | | | | | | |
| Black | | 0.87 | 0.30, 2.38 | 0.8 | 2.11 | 0.59, 7.60 | 0.2 |
| Asian | | 2.57 | 1.02, 6.57 | 0.046 | 2.94 | 0.94, 9.78 | 0.068 |
| Other | | 0.25 | 0.01, 1.45 | 0.2 | 0.41 | 0.02, 3.18 | 0.5 |
| BMI | 130 | | | | | | |
| Normal Weight | | | | | | | |
| Overweight or Obese | | 1.90 | 0.87, 4.33 | 0.10 | 3.06 | 1.16, 8.74 | 0.029 |
| Smoking status | 117 | 0.67 | 0.40, 1.08 | 0.12 | | | |
| Any comorbidity | 130 | 1.29 | 0.46, 4.21 | 0.7 | | | |
| Lowest P/F ratio on first ICU day | 126 | 0.91 | 0.84, 0.97 | 0.006 | 0.90 | 0.81, 0.98 | 0.016 |
| pH at time of lowest P/F ratio | 126 | 0.01 | 0.00, 1.09 | 0.058 | | | |
| $PaCO_2$ at time of lowest P/F ratio | 126 | 1.40 | 1.02, 1.95 | 0.041 | 1.52 | 1.01, 2.39 | 0.050 |

[1]OR = Odds Ratio, CI = Confidence Interval

Table 6. Relationships between factors during ICU admission and outcome.

| Characteristic | N | Univariable | | | Multivariable | | |
|---|---|---|---|---|---|---|---|
| | | OR[1] | 95% CI[1] | p-value | OR[1] | 95% CI[1] | p-value |
| Age on admission | 130 | 1.12 | 1.07, 1.18 | <0.001 | 1.17 | 1.09, 1.27 | <0.001 |
| Lowest P/F ratio during ICU admission | 127 | 0.80 | 0.71, 0.88 | <0.001 | 0.82 | 0.67, 0.98 | 0.036 |
| Lowest pH ratio during ICU admission | 127 | 0.00 | 0.00, 0.01 | <0.001 | | | |
| Highest $PaCO_2$ during ICU admission | 127 | 2.00 | 1.43, 2.89 | <0.001 | 1.30 | 0.74, 2.34 | 0.4 |
| Lowest $PaO_2$ during ICU admission | 127 | 0.70 | 0.45, 1.06 | 0.11 | | | |
| Highest PEEP during ICU admission | 117 | 1.15 | 0.99, 1.35 | 0.072 | 0.94 | 0.73, 1.20 | 0.6 |
| Highest PIP during ICU admission | 112 | 1.15 | 1.04, 1.29 | 0.010 | 1.05 | 0.87, 1.27 | 0.6 |
| Highest noradrenaline dose during ICU admission | 128 | 32.2 | 3.97, 341 | 0.002 | 33.0 | 1.61, 860 | 0.027 |
| Highest temperature during ICU admission | 128 | 1.12 | 0.97, 1.70 | 0.3 | | | |
| Intubated | 130 | 2.46 | 0.76, 11.0 | 0.2 | | | |
| Neuromuscular blockade | 130 | 5.82 | 2.36, 16.6 | <0.001 | 6.48 | 0.96, 53.4 | 0.064 |
| Prone position ventilation | 130 | 3.37 | 1.54, 7.73 | 0.003 | 0.76 | 0.16, 3.56 | 0.7 |
| PE diagnosed during admission | 73 | 1.28 | 0.48, 3.49 | 0.6 | | | |
| Renal replacement therapy | 130 | 1.66 | 0.74, 3.65 | 0.2 | | | |

[1]OR = Odds Ratio, CI = Confidence Interval

## Discussion

In this prospective observational cohort study, we found that patients admitted to the ICU of a London teaching hospital were mostly male, aged over 60 years and with a high prevalence of comorbidities. A substantial proportion were from ethnic minorities. Patients were critically ill with severe hypoxaemia, almost all received mechanical ventilation, the vast majority required cardiovascular support and there were high rates of renal failure and thromboembolic complications.

Our study is the one of the two largest single centre analyses, published to date, describing cohorts of critically ill patients with COVID-19 in Europe [7, 17]. Larsson and colleagues [7] reported on the characteristics and outcomes of 260 patients admitted to ICU at the Karolinska Institute in Stockholm, although almost one quarter of patients did not have a completed ICU episode at the time of analysis and their study lacked detailed information on physiological variables and treatments received on ICU. The UK Intensive Care National Audit and Research Centre (ICNARC) has published regular reports throughout the pandemic [16], summarised in a recent peer-reviewed publication [18]. These analyses have been limited to physiological data from the first 24 hours of admission and have lacked detailed information on symptoms and disease-specific therapies received on ICU, such as prone position ventilation. Other reports from the UK include a study using administrative data to evaluate differences in mortality between hospitals [19], a study focussing on the use of risk scores to predict outcome in patients admitted to ICU with COVID-19 [20], an analysis of the demographic characteristics of a small cohort of patients admitted to ICU [21], and a highly selected case-control series [22] published on the preprint server medRxiv.org. A large retrospective, telephone-based cohort study from Lombardy, Italy [6] conducted a comprehensive analysis of comorbidities, respiratory physiology and the use of prone position ventilation although their study was again limited to data from the first 24 hours of admission and only 42% of patients had a completed ICU episode at the time of publication. Two recent systematic reviews [17, 23] have summarised the available data from cohort studies around the world.

The demographic characteristics of our patient cohort–almost three quarters male, more than half overweight or obese, more than 40% from ethnic minorities, more than half aged over 60 years–closely mirror those seen in other studies [6–8, 10, 18]. The prevalence of comorbidities was high, with only 16% reporting no past medical history. Data from all ICUs in England, Wales, and Northern Ireland, as reported by ICNARC [18], found that 70% of patients were male, 74% were overweight or obese, and 36% were from ethnic minorities, with a median age of 60. These findings closely mirror those seen at our institution. Large cohort studies from New York City [8], Atlanta [10], Lombardy [6] and Stockholm [7] reached similar conclusions. It is noteworthy that raised BMI was associated with increased mortality in this current study, even after adjustment for possible confounding factors in a multivariable logistic regression model, with Asian ethnicity almost reaching the threshold for statistical significance. The proportion of patients of Asian or Black ethnicity admitted to our ICU with COVID-19 is much higher than would be expected given the makeup of the local population [24]. Further research is urgently required to understand the mechanisms underpinning these observations, which have been consistently noted in a number of studies [18, 21, 25–27].

The patients admitted to our ICU had severe hypoxaemic respiratory failure. Almost all patients required intubation and mechanical ventilation, in keeping with the New York [8], Atlanta [10], Lombardy [6] and Stockholm [7] cohorts, although the requirement for invasive ventilation was much higher than reported in Chinese studies [4, 5, 28–30]. This may reflect differences in the use of non-invasive ventilation between countries and the settings within the hospital where these therapies are provided, and highlights the importance of considering regional data when planning for potential future waves of the pandemic.

More than half of the intubated patients on our ICU required neuromuscular blockade and/or prone position ventilation; the use of these therapies was much higher than reported in early studies [6, 8], although was similar to the findings of a more recent report from Norway [31]. The association between neuromuscular blockade and prone position ventilation and death in univariable models is likely to reflect confounding by indication, whereby the most severely unwell patients, with refractory hypoxaemia, were more likely to be receive neuromuscular blockade and/or be placed in the prone position. Although there is high quality evidence of a mortality benefit from prone position ventilation in patients with ARDS [32], it is unclear whether this extends to patients with COVID-19. In the event of another wave of infection further studies are required to address this important question. Furthermore, the intense resource commitment required to safely ventilate large numbers of patients in the prone position should be borne in mind when planning for any future outbreaks of COVID-19 infection.

The majority of patients admitted to our ICU had multiorgan failure, defined as the requirement for at least two of respiratory, cardiovascular or renal support, with almost three quarters requiring at least one vasoactive drug and more than one quarter requiring renal replacement therapy. The high prevalence of acute kidney injury in patients with COVID-19 has been widely reported [4, 8, 18, 33] and requires urgent further investigation to understand the mechanisms involved. Similarly, high rates of renal replacement therapy have been reported in other UK ICUs [18] and in cohorts from New York [8], Dublin [34] and Stockholm [7] but not China [30, 35]. The requirement for multiorgan support must be borne in mind when it comes to planning for further waves of infection; it is clear that a focus on ICU ventilators, for example, will not be sufficient. Adequate plans to provide vasopressor and inotropic drugs by infusion, along with renal replacement therapy, must be made.

A greater than expected number of patients in our cohort were diagnosed with a PE and more than one third of these had CT evidence of right ventricular dysfunction. Thromboembolic complications have been widely reported in patients with COVID-19 [36, 37], including

in patients admitted to ICU [38]. Further work is required to understand the role of screening for PEs in patients admitted to ICU with COVID-19, and determine the most effective treatment strategy.

Strengths of our study include its relatively large sample size, the complete ascertainment of all patients admitted to ICU with COVID-19 at our institution, the prospective design using a standardised, internationally recognised data collection tool, the granular and highly curated dataset collected on each patient through manual chart review, and follow-up for at least 28 days in every patient.

Our study had a number of limitations. Like all observational designs it is subject to confounding and associations between exposures and outcomes should not be interpreted as causal relationships. The population admitted to ICU was a subset of those presenting to and admitted to hospital. Upstream triage of patients and the criteria used to identify those patients requiring (and suitable for) ICU admission will have affected the composition of our cohort and potentially the relationships between exposures and outcomes. The criteria used for ICU admission are likely to have varied between institutions and at different time points during the pandemic. As such, the findings in our cohort may differ from those in other studies, and they may not represent the entire population of patients severely ill with COVID-19. The lack of data concerning the population admitted to our hospital but not our ICU limits our ability to explore this inclusion bias is more detail, although it is reassuring to note the similarities between our findings and those of other cohorts. Physiological data were recorded on paper charts and as such only a small subset of observations could be digitised for analysis. A number of patients were transferred out of the hospital for logistical reasons and we were unable to gather information beyond their survival status once they left our ICU. This is likely to have biased in favour of increased mortality since the most stable patients were chosen for transfer. Although patients were transferred to our hospital from across London the local population is not representative of London as a whole in terms of its ethnic and sociodemographic makeup. We have not controlled for multiple analyses and the possibility of type I error cannot be excluded.

## Conclusions

In this large cohort of hypoxaemic critically ill patients admitted to an ICU in London with COVID-19, we demonstrated that age, obesity and degree of hypoxaemia were independently associated with increased odds of death. There was a strong signal towards an association between Asian ethnicity and death in univariable analyses. Multiple organ failure requiring support was common as was the diagnosis of PE. In the event of further waves of this pandemic in the UK, sufficient plans must be in place to cope with this expected pattern of disease and studies must be ready to explore the links between obesity, ethnicity and survival.

## Acknowledgments

We are grateful to all the doctors, nurses, allied health professionals and support staff who worked so hard to provide high quality care throughout this challenging period. We are particularly grateful to Peggy Tsang, Caroline Butler, Fiona Xang, Sharon Yip and Zhangqi Zhao who collected the Intensive Care National Audit and Research Centre (ICNARC) data.

## Author Contributions

**Conceptualization:** Ross J. Thomson, Jennifer Hunter, Jonathan Dutton, Kulwant Dhadwal, Daniel Martin.

**Data curation:** Ross J. Thomson, Jennifer Hunter, Jonathan Dutton, James Schneider, Maryam Khosravi, Alisha Casement.

**Formal analysis:** Ross J. Thomson, Jennifer Hunter.

**Investigation:** Ross J. Thomson, Jennifer Hunter, Jonathan Dutton, James Schneider, Maryam Khosravi, Alisha Casement.

**Methodology:** Ross J. Thomson, Jennifer Hunter, Jonathan Dutton, Daniel Martin.

**Project administration:** Ross J. Thomson, Kulwant Dhadwal.

**Supervision:** Kulwant Dhadwal, Daniel Martin.

**Writing – original draft:** Ross J. Thomson.

**Writing – review & editing:** Ross J. Thomson, Jennifer Hunter, Jonathan Dutton, James Schneider, Maryam Khosravi, Alisha Casement, Kulwant Dhadwal, Daniel Martin.

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
