## [Decision Letter · Decision Letter 0]

2 Nov 2020

PONE-D-20-24459

Clinical characteristics and outcomes of critically ill patients with COVID-19 admitted to an intensive care unit in London: a prospective observational cohort study

PLOS ONE

Dear Dr. Thomson,

Thank you for submitting your manuscript to PLOS ONE. After careful consideration, we feel that it has merit but does not fully meet PLOS ONE’s publication criteria as it currently stands. Therefore, we invite you to submit a revised version of the manuscript that addresses the points raised during the review process.

We suggest the authors to better elucidate inclusion and exclusion criteria and eventually related bias. The peculiarity of these findings should be better described in the discussion section. 

We look forward to receiving your revised manuscript.

Kind regards,

Chiara Lazzeri

Academic Editor

PLOS ONE

Journal Requirements:

2. Please clarify the name of the IRB that  classified the study as a non-interventional service evaluation and waived participant consent.

3.Thank you for stating the following in the Competing Interests section:

[I have read the journal's policy and the authors of this manuscript have the following competing interests:

DM has received consultancy and lecture fees from Edwards Lifesciences and Siemens Healthineers; none of the other authors have any conflicts of interest to report.].

4.We note that you have indicated that data from this study are available upon request. PLOS only allows data to be available upon request if there are legal or ethical restrictions on sharing data publicly. For information on unacceptable data access restrictions, please see http://journals.plos.org/plosone/s/data-availability#loc-unacceptable-data-access-restrictions.

Reviewers' comments:

Reviewer's Responses to Questions

**Comments to the Author**

1. Is the manuscript technically sound, and do the data support the conclusions?

Reviewer #1: Yes

2. Has the statistical analysis been performed appropriately and rigorously? 

Reviewer #1: Yes

3. Have the authors made all data underlying the findings in their manuscript fully available?

Reviewer #1: No

4. Is the manuscript presented in an intelligible fashion and written in standard English?

Reviewer #1: Yes

5. Review Comments to the Author

Reviewer #1: Interesting & well presented case series which would be informative to clinical practice .

Detailed & granular baseline information and complete follow up across multiple variables

Very good figures and tables to illustrate the results . I agree with the methodology for constructing the 2 models , and with the statistical analysis plan .

As the authors comment in their discussion - a case series such as this is vulnerable to bias . Several are discussed. Inclusion bias is important - the study aims to characterise a "typical" population who develop severe COVID 19 , and describe their clinical course, treatment and prognostic factors. Clearly the "ICU" population is skewed towards those patients who were hospitalised with severe COVID and were then referred for Critical Care treatment. It is likely that there were several patients admitted to your hospital who had severe COVID , and who died from COVID who were not referred for ICU care ( but might have been , had they attended a different hospital , or a healthcare system in a different country.

Or who were potentially not referred due to upstream "triaging " decisions for example by ED or acute medical teams deciding not to refer so as to "protect" ICU beds . Many of these pts may have been just as "sick" in terms of dependency , but don`t appear in your case series

For context ie to understand how the ICU population in your hospital compares with the population of patients admitted with COVID -can you provide some basic information about the overall admissions ( number of patients admitted, median age and IQR , mortality rate, LOS in hospital )

and possibly also similar descriptors for the cohort of pts who died from or with COVID -19 following hospital admission, but were either never referred, or were declined for ICU admission ?

It may be difficult or impossible to obtain granular information about such patients - it should at least be mentioned as a key limitation ?

Gary Minto, Plymouth UK

Minor : p 13 line 294 Of the 156 total admissions to ICU with COVID-19, 38 (24%) patients died on ICU, 23 (15%) patients

213 were transferred out to other hospitals, 92 (59%) patients were discharged alive from ICU and the

214 remaining 3 (2%) patients were still on ICU - SHOULD SAY SOMETHING LIKE "at the time of follow up "

6. PLOS authors have the option to publish the peer review history of their article (what does this mean?). If published, this will include your full peer review and any attached files.

Reviewer #1: **Yes: **Gary Minto, Plymouth UK

---

## [Author Response · Author response to Decision Letter 0]

22 Nov 2020

Responses to Academic Editor

We have actioned this point in the revised submission.

2. Please clarify the name of the IRB that classified the study as a non-interventional service evaluation and waived participant consent.

The legal framework governing medical research in the United Kingdom does not require institutional review board or ethics committee approval for studies that meet the definition of non-interventional service evaluations, nor does it require an IRB to decide that a study meets this definition. The study was registered with the Institutional Research Office of the Royal Free London NHS Foundation Trust as a non-interventional service evaluation. Even if this were not the case, and participant consent would ordinarily be required, the UK government has instituted a consent waiver for the purposes of COVID-19 research (notice under regulation 3(4) of the Health Service (Control of Patient Information) Regulations 2002).

3. Please see our revised competing interests statement below:

I have read the journal's policy and the authors of this manuscript have the following competing interests:

DM has received consultancy and lecture fees from Edwards Lifesciences and Siemens Healthineers; none of the other authors have any conflicts of interest to report. This does not alter our adherence to PLOS ONE policies on sharing data and materials.

4. Data availability

This study utilised data collected from patients admitted to hospital with COVID-19. This confidential patient data is protected by ethical considerations (patients have a right to confidentiality), the professional obligations of the researchers (who are bound by the policies of the General Medical Council, the statutory body that regulates doctors in the UK) and by UK law. Unrestricted sharing of the data underpinning the study is therefore not possible on ethical and legal grounds. We do not believe it is possible to sufficiently anonymise the data while maintaining its usefulness for research. The dataset is extremely granular and concerns a relatively small number of individuals. We are therefore concerned that individuals would be potentially identifiable even if overt identifiers (name, date of birth, etc.) were removed. Furthermore, repurposing the data so that it could be shared would not be compatible with the legal basis (service evaluation) under which we have conducted our study.

We would be happy to share the data underpinning the study to facilitate replication of our findings or further research, but this needs to be done in a manner compatible with UK law. This would involve a group of researchers obtaining ethical and research governance approvals and their institution entering into a data sharing agreement with ours. We have provided contact details for our Institutional Research Office in our data availability statement.

We have updated our data sharing agreement:

The data that support the findings of this study are not publicly available out of consideration for patient confidentiality. On account of the highly granular nature of the clinical dataset it is not possible to anonymise it sufficiently to prevent reidentification of participants without jeopardising its usefulness for research. The Institutional Research Office (rf.randd@nhs.net) can facilitate access to the data for appropriately qualified individuals provided that the correct ethical permissions and data sharing agreements are in place.

Responses to Reviewer

Thank you for your detailed and careful consideration of our manuscript.

1. Inclusion bias

We entirely agree with the reviewer’s comments regarding inclusion bias. The subset of patients admitted to ICU represents a group who were critically unwell with COVID-19, and who were thought likely to benefit from admission to critical care.

As an organisation we devised a guideline on the care of patients with COVID-19. This was developed jointly by physicians from the ICU, acute medicine and the emergency department. We employed a number of objective criteria to assess illness severity (and hence need for ICU admission). We used the World Health Organisation’s definition of “critical” COVID-19 (the presence of ARDS, sepsis or septic shock) to identify those patients who definitely required admission to critical care. We used the WHO’s definition of “severe” COVID-19 (respiratory rate > 30 breaths/min; severe respiratory distress; or SpO2 < 90% on room air) to identify those at high risk of deterioration who required close monitoring. Inevitably there were patients admitted to ICU who did not meet these criteria, for example because they had comorbidities (e.g. organ transplantation) or other conditions (e.g. acute renal failure) that necessitated treatment in a critical care setting.

We used the guidance from the National Institutes of Health and Care Excellence to assess patients’ likelihood of benefitting from ICU admission. This involved calculating each patient’s Clinical Frailty Score, with the frailest patients being least likely to survive admission to critical care.

Through the use of objective criteria for illness severity requiring ICU admission, and frailty that would render ICU admission futile, we attempted to standardise decision making. The reviewer is correct to point out that an element of subjectivity is inevitable, and that thresholds for critical care admission will have changed over time and with our understanding of the disease. It is therefore likely that the mix of patients in ICU will vary between institutions, countries and even within the same institution over the stages of the pandemic.

We have updated the Methods section to explain our policy for critical care admission, and the Discussion to better explain the limitations imposed by inclusion bias. We have also placed ICU admissions in context by providing data on the total number of patients admitted to the hospital during the study period and their average age. Unfortunately our institution employs a paper-based health record and the administrative data recorded electronically is very limited, restricting our ability to provide detail of the cohort of patients admitted to hospital but not requiring ICU admission.

2. Minor: p 13 line 294 Of the 156 total admissions to ICU with COVID-19, 38 (24%) patients died on ICU, 23 (15%) patients were transferred out to other hospitals, 92 (59%) patients were discharged alive from ICU and the remaining 3 (2%) patients were still on ICU - SHOULD SAY SOMETHING LIKE "at the time of follow up "

Thank you – we have updated the manuscript to make this clear.

---

## [Editor Report · Decision Letter 1]

30 Nov 2020

Clinical characteristics and outcomes of critically ill patients with COVID-19 admitted to an intensive care unit in London: a prospective observational cohort study

PONE-D-20-24459R1

Dear Dr. Thomson,

We’re pleased to inform you that your manuscript has been judged scientifically suitable for publication and will be formally accepted for publication once it meets all outstanding technical requirements.

Kind regards,

Chiara Lazzeri

Academic Editor

PLOS ONE
---

## [Editor Report · Acceptance letter]

7 Dec 2020

PONE-D-20-24459R1 

Clinical characteristics and outcomes of critically ill patients with COVID-19 admitted to an intensive care unit in London: a prospective observational cohort study 

Dear Dr. Thomson:

I'm pleased to inform you that your manuscript has been deemed suitable for publication in PLOS ONE. Congratulations! Your manuscript is now with our production department. 

Kind regards, 

on behalf of

Dr. Chiara Lazzeri 

Academic Editor

PLOS ONE